# Study on the Performance of Acrylic Polyurethane for the Protection of Handwriting on Paper Relics

**Tan Mao** [1,2,*], **Xufeng Li** [1], **Xiaoting Shi** [1], **Ying Hu** [1], **Junyan Zha** [1], **Xueke Luo** [1,2] **and Youliang Cheng** [2]

1   College of Mechanical and Material Engineering, North China University of Technology, Beijing 100144, China
2   College of Mechanical and Precision Instrument Engineering, Xi'an University of Technology, Xi'an 710048, China
*   Correspondence: t_maoncut@163.com

**Abstract:** There are many important cultural relics in China, including many paper relics. Therefore, it is particularly important to find methods of protecting the handwriting in paper relics. This study focuses on the effects acrylic polyurethane materials have on preserving handwriting in paper relics by examining the color change and diffusion of handwriting before and after reinforcement treatments on handwriting materials, including ink, carbon ink, blue-black ink, pure blue ink, red ink, and pencil. The acrylic polyurethane materials have a certain degree of stability and enhance the handwriting of paper relics. We analyzed the acrylic polyurethane reinforcement material in terms of acid resistance, alkali resistance, oxidation resistance, dry heat aging resistance, and durability. The experimental results show that the optimal reinforcement concentration is within 15% for ink, 10% for carbon ink, 20% for blue-black ink, 15% to 20% for pure blue ink, and 3% for red ink and pencil. Finally, the surface of the reinforced paper and handwriting were observed through laser microscopy, SEM, EDS, and infrared spectroscopy and the protective effect of acrylic polyurethane reinforcement material on handwriting was determined.

**Keywords:** paper cultural relics; acrylic polyurethane; writing materials; reinforce the protection





## 1. Introduction

Paper relics are the historical symbols of each era and have great artistic and research value. Over time, paper relics are easily damaged. Thus, studying the protection and restoration of paper relics is an essential field of cultural relics research [1].

In recent years, natural polymer and resin reinforcement have been the primary methods for protecting cultural relics. Acrylic resin is one of them. Acrylic resins have good film-forming, weathering, transparency, and stain resistance, and are widely used to protect relics of different materials [2] such as pottery [3], glass [4], frescoes [5], textiles [6], paper [7], and wood [8]. However, the preparation requirements for conventional acrylic resin are complicated, including the high temperature of film formation and the demand for greater hardness. This needs to be improved in the field of cultural relic protection [9]. Therefore, many scholars have started to modify and further improve the performance of this material. Zhang et al. synthesized a novel carboxymethyl cellulose acrylate OVPOSS graft copolymer for paper protection that significantly increased the mechanical strength of the paper [10]. Chen et al. prepared modified carboxymethyl cellulose/Si/polyacrylate protective materials using acrylic resin and modified carboxymethyl cellulose, which improved the mechanical strength and weather resistance of the paper [11]. Qiao et al. used poly (methyl methacrylate and methyl acrylate co-styrene) to strengthen the fragile paper fibers, which significantly improved the mechanical properties and aging resistance of papers [12]. Xu et al. have prepared a series of novel microcrystalline cellulose/$TiO_2$/fluorine/styrene-acrylate coatings using modified microcrystalline cellulose, modified nano$TiO_2$, dodecafluoroheptyl methacrylate, and several acrylate monomers. The protective layer formed by

this material on paper has good hydrophobicity, oil repellency, mechanical strength, aging resistance, and reversibility [13]. In addition, the self-healing acrylic copolymer composite coatings invented by the team have good anti-aging and self-healing performance [14].

So far, most of the research on paper cultural relic protection focuses on paper deacidification [15–19] and paper reinforcement. This is also reflected in the above research. Nevertheless, the value of paper cultural relics sometimes lies in something other than the paper carrier itself. The handwriting on the paper is the soul of the existence of paper cultural relics; therefore, whether the handwriting is intact determines the lifespan and value of the paper.

Handwriting may inevitably experience discoloration, diffusion, and other damage during long-term preservation. In the early days of China, substances such as alum water, organic glass solution, and gelatin glycerol were often used to reinforce paper cultural relics [20]. However, these reinforcement materials still have defects, which can cause problems such as handwriting diffusion and paper wrinkles during the reinforcement process. Therefore, it is essential to search for new handwriting reinforcement materials. However, there has been little research conducted on handwriting protection. In China, in 1997, Qu et al. published a research report on the durability testing of handwriting materials, using color difference as the standard for evaluating the durability of handwriting [21]. Lei et al. prepared a protective agent to prevent the diffusion of red and blue ink handwriting by adjusting the mass ratio of chitosan, acetic acid, hydrogen peroxide, and water [22]. Qu explored the effects of oxalic acid/ethanol and barium hydroxide/methanol protective solution, aqueous fluorocarbon lotion, and gelatin glycerin solution on the durability of handwriting [23]. Internationally, researchers are more focused on exploring the effects of pigment composition [24–27] and external factors on handwriting [28–33].

The above research indicates that acrylic acid has broad prospects in paper cultural relic protection. Therefore, we selected polyisocyanate (HDI trimer) with -NCO groups and hydroxyl acrylic resin (MR7323E) with -OH groups mixed reaction to prepare acrylic polyurethane materials. The color difference value ($\Delta E^*$) is used as the criterion for judging the reinforcement effect to explore the applicability of acrylic polyurethane material in paper heritage handwriting and provide supporting data.

## 2. Materials and Methods

### 2.1. Instruments and Materials

2.1.1. Experimental Instruments

Electronic balance (BSM220.4; accuracy: 0.0001 g; Shanghai Zhuojing Electronic Technology Co., Ltd.; Shanghai, China); color difference meter (SC-10; accuracy: $\Delta E \leq 0.03$; Shenzhen Sanenshi Technology Co., Ltd.; Shenzhen, China); electric hot air drum drying oven (GZX-9246MBE; Accuracy: $\pm 1\,^{\circ}C$ (105 $^{\circ}C$); Shanghai Boxun Medical Biological Instrument Co., Ltd.; Shanghai, China); infrared spectrometer (Thermo Scientific; Nicolet 380; accuracy: 4000–450 cm$^{-1}$, $\pm 0.5$ cm$^{-1}$; Thermo Fisher Scientific; Waltham, MA, USA); scanning electron microscopy (SU8100; Hitachi; Tokyo, Japan); 3D measurement laser microscope (OLS4100; Olympus Corporation; Tokyo, Japan); transmission electron microscope (JEOL 2100F; JEOL; Tokyo, Japan); paper cutter (BEFON; Beijing, China); and brush (HANDSKIT; Shenzhen, China).

2.1.2. Experimental Materials

Sodium hydroxide (NaOH, 0.01 mol/L); hydrochloric acid (HCl, 0.01 mol/L); hydrogen peroxide (1% $H_2O_2$); butyl acetate ($C_6H_{12}O_2$, $\geq$99.0% purity); HDI trimer (N3390,-NCO 19.6 $\pm$ 0.3%, Bayer); hydroxyacrylic resin (MR7323E,- OH 33%, KDD Functional Material); ink (Beijing Yidege Ink Co., Ltd., Beijing, China); carbon ink, blue-black ink, pure blue ink, red ink (Shanghai Fine Cultural Articles Co., Ltd., Shanghai, China); China drawing 2B pencil (China's First Pencil Bengbu Co., Ltd., Bengbu, China); rice paper (Anhui Jing County Huiji Rice Paper Co., Ltd., Bengbu, China); and Xuan paper (a paper made using ancient Chinese methods, Jing County, Bengbu, China).

The main components of the ink, carbon ink, and pencil are inorganic pigments. The main components of blue-black ink are organic pigments and the main components of pure blue and red ink are dyes [34]. Using these experimental materials, we imitated the different components of handwriting materials in paper relics, finally achieving the experimental purpose.

### 2.2. Experimental Methods

#### 2.2.1. Sample Preparation

For a more realistic simulation of heritage paper, we chose the handmade Xuan paper made using the ancient method in Jingxian County, Anhui Province to be the experimental carrier. We cut the paper into 5 cm × 5 cm pieces. We used the six handwriting materials to coat it with color blocks in an area of 3 cm × 3 cm. We left it to dry at room temperature and observed the color difference and diffusion before and after reinforcement. As shown in Figure 1.

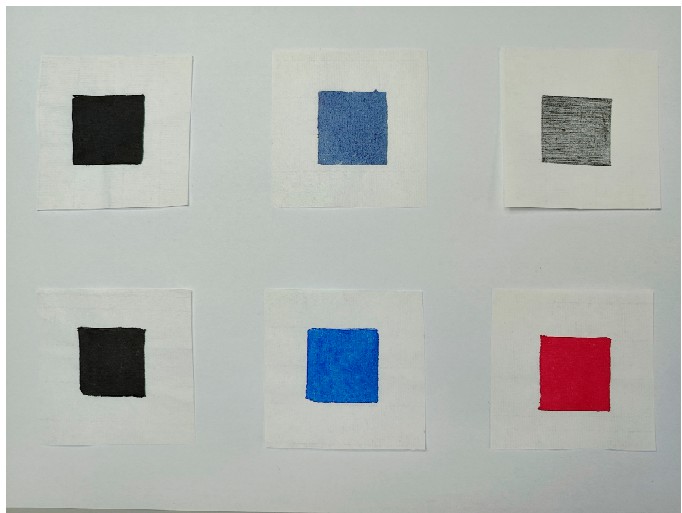

**Figure 1.** Paper sample.

#### 2.2.2. Hardening

This experiment adopts the brushing method, where a brush is dipped in a reinforcing agent and applied to the handwriting color block of the sample. After brushing, it is immediately blown dry to prevent the rapid diffusion of handwriting. The advantage of this method is that it is easy to operate and control, and suitable for small-area reinforcement.

#### 2.2.3. Performance Test

According to DA/T 16-1995 (Chinese National Standard), the experimental design steps are as follows:

1. Acid resistance: inject 60 mL 0.01 mol/L HCL solution into a clean beaker and immerse the sample in the solution for 1 h. Then, remove and wash 3 times with distilled water and dry in the shade without sunlight for testing.
2. Alkali resistance: similar to step 1, change the solution to 0.01 mol/L NaOH.
3. Oxidation resistance: similar to step 1, change the solution to 1% $H_2O_2$.
4. Dry heat aging resistance: put the sample in a dry heat aging box with a temperature (T) of $105 \pm 5\,°C$ for 72 h. Then, take it out and cool it to room temperature. It is worth noting that it must be kept away from light.

#### 2.2.4. Color Difference Detection

In order to quantitatively express the color change of the experimental samples [35], we adopted the CIELAB standard chroma table color system of the International Commission on Illumination to evaluate the color change. The chromatic difference ΔE* of two colors is

the geometric distance between their two positions in CIE1976 (L*, a *, b *) color space [36], which is calculated by the following formula:

$\Delta E^* = [(\Delta L^*)^2 + (\Delta a^*)^2 + (\Delta b^*)^2]^{1/2}$, among them:

$\Delta L^* = L^*$ Sample—$L^*$ Standard (light/dark difference)

$\Delta a^* = a^*$ Sample—$a^*$ Standard (red/green difference)

$\Delta b^* = b^*$ Sample—$b^*$ Standard (yellow/blue difference)

The smaller the value of $\Delta E^*$, the less obvious the color change. This demonstrates the reinforcement effect of reinforcement materials. We measured the color of the handwriting materials before and after accelerated aging to obtain the outstanding difference ($\Delta E^*$) and combined these into a color change control table, shown in Table 1. A remarkable characteristic is that the test data from each experiment are singular and quick [37]. (According to the Chinese color system standard GB/T15608-2006, $\Delta E^* \leq 1.5$ is the color tolerance.)

**Table 1.** Comparison table of color difference value and color change.

| Color Difference Value (NBS) | Color Change |
| --- | --- |
| 0~0.5 | Change of mark |
| 0.5~1.5 | Slight variation |
| 1.5~3.0 | Visible change |
| 3.0~6.0 | Significant change |
| 6.0~12.0 | Change is big |
| More than 12.0 | Change is very big |

## 3. Results and Discussion

### 3.1. Solution Preparation and Validation

In order to ensure that the resulting acrylic polyurethane solution still has a sufficient amount of -NCO to combine with the -OH in the paper fiber, the HDI trimer and hydroxy acrylic acid were mixed in a butyl acetate solution in the ratio of -NCO: -OH = 1.5, with constant stirring until the solution was clear and without transparent precipitation (the concentrations in this test are all mass percent concentrations). The reaction process is shown in (1) in Figure 2. The reaction in (2) can be carried out by coating the reinforcement solution on the handwriting surface.

R₁:Acrylic resin、CELL:The paper fiber

**Figure 2.** Chemical reaction formula.

After continuous stirring until the solution is clear and without transparent precipitation, the acrylic polyurethane solution can be obtained. The reaction process is shown in (1) in Figure 2.

To verify that the resulting solution is as expected in (1) in Figure 2, experiments on the infrared spectra were performed, as shown in Figure 3.

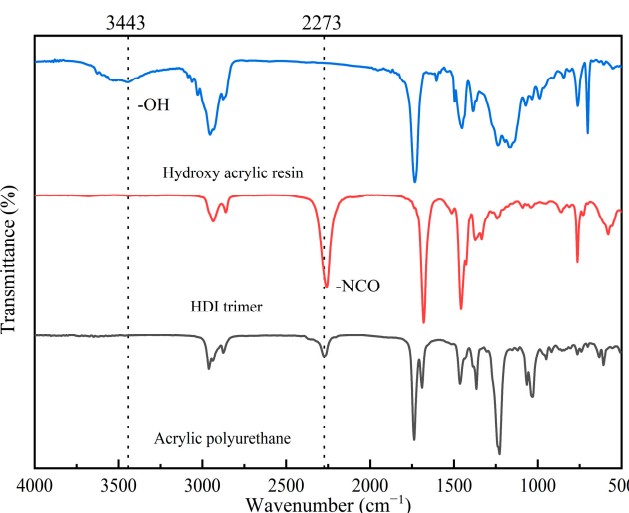

**Figure 3.** Infrared spectroscopy detection map.

It can be seen that the -OH group of peak 3443 cm$^{-1}$ in the spectrum of hydroxy acrylic acid has completely disappeared in the spectrum of acrylic polyurethane. This shows that the hydroxyl acrylic acid reacts fully with the HDI trimer, completing the chemical reaction shown in (1) in Figure 2. The -NCO group at peak 2273 cm$^{-1}$ of the HDI trimer spectrum was still retained in the acrylic polyurethane spectrum, indicating that the synthesized acrylic polyurethane solution retained the ability to bond with paper and the prepared reinforcement solution was the same as expected.

Microstructure of Acrylic Polyurethane Materials

The TEM image of acrylic polyurethane is shown in Figure 4. Figure 4 shows that the colloidal particles are circular, indicating that the shell structure of the coating is approximately spherical, which may be related to the symmetrical structure of the HDI trimer.

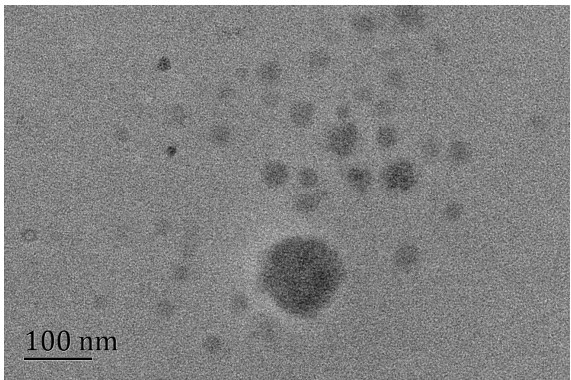

**Figure 4.** TEM image of acrylic polyurethane.

### 3.2. Confirmation of Concentration Range and Elimination of Solvent Influence

Observe the change in handwriting after reinforcement with different concentrations of acrylic polyurethane solutions in the early stage of the experiment. When the concentration of reinforcer exceeds 20% and is lower than 3%, the reinforcement effect is not good. Finally, 3%, 5%, 10%, 15%, and 20% were selected as the uniform test concentrations for this experiment.

In order to eliminate the influence of the solvent on the color difference of paper, we used butyl acetate to coat blank samples and various color handwriting samples. The experimental results show that the average color difference of the blank sample is 0.53, the

average color difference of the handwriting sample is 0.42, and the paper itself is 0.33. The color difference change can be ignored.

### 3.3. Handwriting Changes after Hardening

The self-made acrylic polyurethane solution with different concentrations was smeared on the sample's surface, and the changes in handwriting were observed and recorded. The changes are shown in Table 2. According to GBT 32004-2015 (China National Standard), handwriting diffusion was observed and graded with an 8-fold magnifying glass or similar magnifying tools. In all the following tables, the diffusion phenomenon is represented by these symbols: no noticeable change in handwriting (—); tiny (tiny); mild (+), obvious (++), severe (+++).

**Table 2.** Handwriting diffusion after reinforcement with acrylic polyurethane solution of different concentrations.

| Solution Concentration | Paper Color Difference | Ink | Carbon Ink | Blue-Black Ink | Pure Blue Ink | Red Ink | Pencil |
|---|---|---|---|---|---|---|---|
| 3% concentration | 0.59 | — | — | — | — | — | — |
| 5% concentration | 1.14 | — | — | — | — | — | — |
| 10% concentration | 1.21 | — | — | — | — | — | — |
| 15% concentration | 1.79 | — | — | — | tiny | tiny | — |
| 20% concentration | 2.5 | — | — | — | tiny | tiny | — |

As we can see from Table 2, with the increase in the concentration of the reinforcing solution, the $\Delta E^*$ of the carrier paper gradually increases. From the perspective of the diffusion phenomenon, except for pure blue ink and red ink, other writing materials do not experience diffusion, and pure blue ink and red ink experience only a very slight diffusion phenomenon at 15% and 20% reinforcement concentrations. From the change in handwriting color before and after reinforcement, the $\Delta E^*$ of each writing material changes in disorder according to the increase in reinforcement concentration (as shown in Figure 5). For all handwriting materials, the color difference change of ink is relatively stable. The color difference value is basically within 1.5 in concentrations between 3% and 10%. However, the reinforcing effect of the reinforcing agent on the pencil is not ideal, and all of the $\Delta E^*$ values are above 3. It can be seen that there are apparent reinforcing marks on pencil handwriting, which may be related to the film formation of reinforcing materials on handwriting materials. The $\Delta E^*$ between carbon ink and red ink shows an upward trend with increased concentrations of reinforcer, and the reinforcing effect on carbon ink is better. Both blue-black ink and pure blue ink have a peak color difference at 10% concentration. The $\Delta E^*$ decreases before and after the peak value, while the $\Delta E^*$ of pure blue ink changes smoothly and the reinforcement effect is good. Therefore, it can be seen from the chart that the concentration of suitable reinforcement agents for different writing materials is also different. The applicable concentration of ink is within 15%. The applicable concentration of red ink and pencil is 3%, but pencil handwriting still changes significantly. Pure blue ink performs well within ranges between 3% and 20%. Blue-black ink has two relevant concentrations, 3% and 20%, which can be reasonably selected according to the actual reinforcement situation.

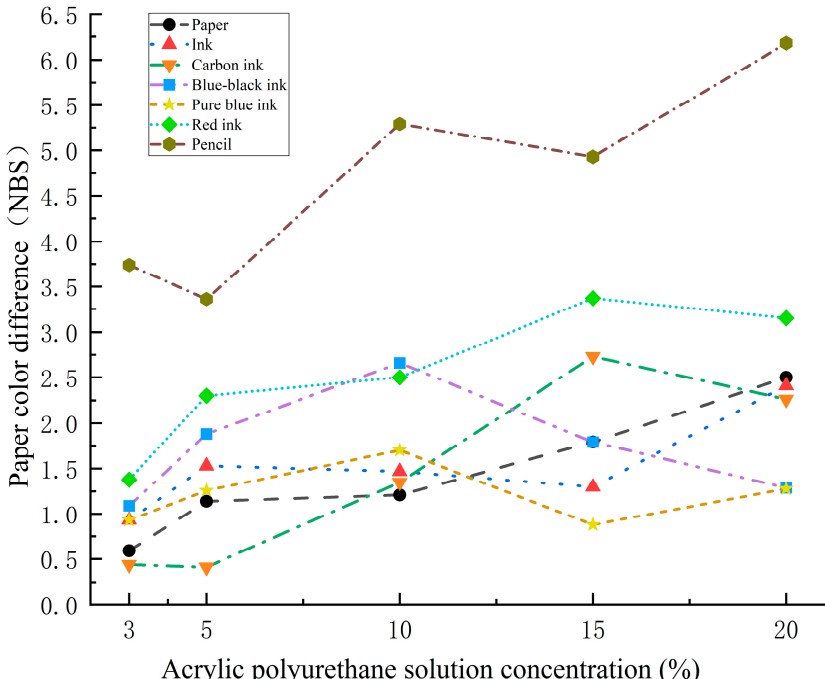

**Figure 5.** Handwriting changes after reinforcement.

### 3.4. Durability Inspection of Handwriting Materials

In order to ensure that the acrylic polyurethane reinforcement has a practical application in the conservation and restoration of paper relics, it is necessary to consider not only whether the reinforcement material has adverse effects on handwriting but also whether it has a good reinforcement effect. The durability of the handwriting material after it has been reinforced is, therefore, a necessary test.

The reinforced handwriting sample is used as the standard sample. The handwriting specimens were subjected to acid, alkali, oxidation, and dry heat aging tests and observed for diffusion and ΔE*. Based on the experimental results, it can be found that:

(1) Acid resistance: the ink and pencil did not diffuse in the experimental or matched groups. The ΔE* of the ink was relatively small and the experimental group data were smaller than the matched group, with a minimum color difference of 0.18 at 10%. The ΔE* of the pencil irregularly changes, showing a higher color difference than the matched group at 3% and 15% concentrations. Therefore, the acrylic polyurethane solution was unsatisfactory for pencil reinforcement in the acid resistance experiment. There was slight diffusion in the carbon ink matched group; none of the experimental groups showed diffusion, and the ΔE* was better than the matched group. In this experiment, blue-black ink showed a slight diffusion in the experimental and matched groups below 5% concentration of the reinforcing agent and no diffusion above 5%. The difference between the experimental and control groups was significant compared to the 15.19 (ΔE*) in the matched group, showing a good reinforcement effect. The pure blue and red ink have serious diffusion in the matched and experimental groups. The ΔE* data show that the acrylic polyurethane solution still has a reinforcement effect on these two handwriting materials, and the reinforcement effect on red ink is more prominent (as shown in Table 3 and Figure 6).

**Table 3.** Diffusion results of acid resistance experiment on reinforced handwriting materials.

| Writing Material Is Acid Resistant | Matched Group | Concentration | | | | |
|---|---|---|---|---|---|---|
| | | 3% | 5% | 10% | 15% | 20% |
| Ink | — | — | — | — | — | — |
| Carbon ink | + | — | — | — | — | — |
| Blue-black ink | + | + | + | — | — | — |
| Pure blue ink | +++ | +++ | +++ | +++ | +++ | +++ |
| Red ink | +++ | +++ | +++ | +++ | +++ | +++ |
| Pencil | — | — | — | — | — | — |

The diffusion phenomenon is represented by these symbols: no noticeable change in handwriting (—); tiny (tiny); mild (+), obvious (++), severe (+++).

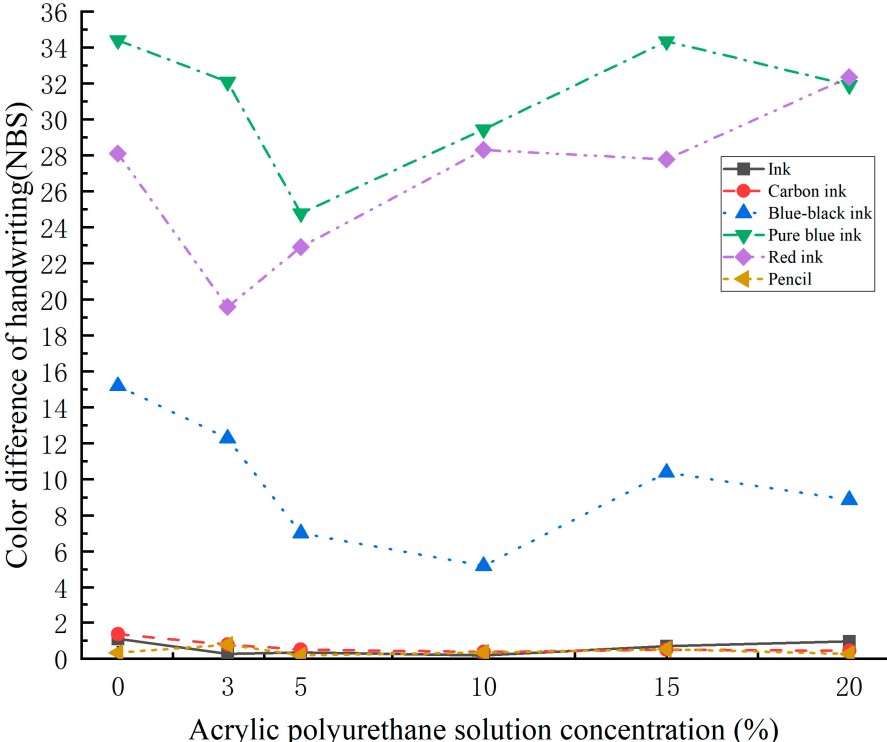

**Figure 6.** Changes in acid resistance of handwriting after reinforcement.

(2) Alkali resistance: there was no diffusion phenomenon between ink and pencil in both the matched and experimental groups. The $\Delta E^*$ of ink was consistent with that of the acid resistance experiment, and both showed increased $\Delta E^*$ under high-concentration reinforcement. However, there was a significant difference in the performance of pencil handwriting materials and acid resistance experiments. The $\Delta E^*$ in the experimental group was much smaller than in the matched group, and the performance was stable at various concentrations, showing good reinforcement performance. The performance of the carbon inks in this experiment was similar to that of the acid resistance experiment. Compared to the acid resistance experiment, the blue-black inks did not have this diffusion at low reinforcement concentrations, and the diffusion phenomenon of pure blue ink was also significantly weaker. The presence of substances that react with NaOH in the red ink caused the discoloration in the experiment, therefore, the $\Delta E^*$ changed more than in the acid resistance test. The $\Delta E^*$ values of pure blue ink, red ink, and blue-black ink in the experimental group were all better than those of the matched group, showing good reinforcement properties (as shown in Table 4 and Figure 7).

**Table 4.** Diffusion results of alkaline resistance experiment on reinforced handwriting materials.

| Writing Material is Alkaline Resistant | Matched Group | Concentration | | | | |
|---|---|---|---|---|---|---|
| | | 3% | 5% | 10% | 15% | 20% |
| Ink | — | — | — | — | — | — |
| Carbon ink | + | — | — | — | — | — |
| Blue-black ink | + | — | — | — | — | — |
| Pure blue ink | ++ | + | + | + | + | + |
| Red ink | +++ | +++ | +++ | +++ | +++ | +++ |
| Pencil | — | — | — | — | — | — |

The diffusion phenomenon is represented by these symbols: no noticeable change in handwriting (—); tiny (tiny); mild (+), obvious (++), severe (+++).

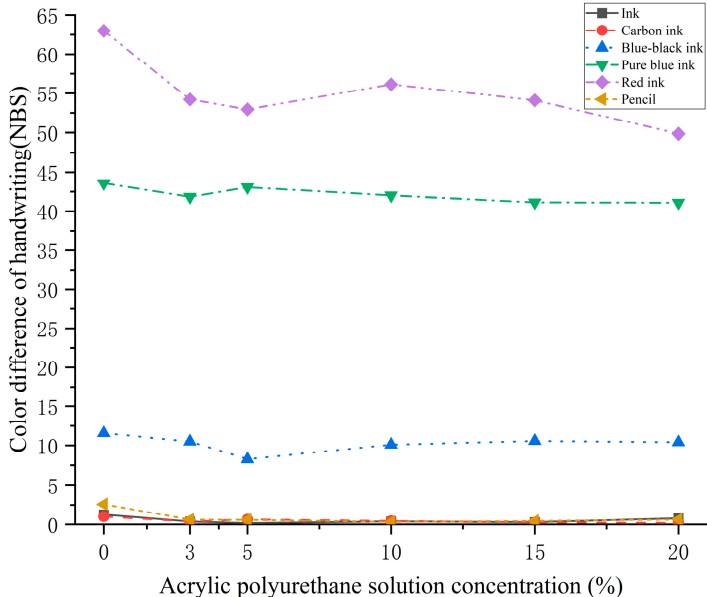

**Figure 7.** Changes in alkaline resistance of handwriting after reinforcement.

(3) Oxidation resistance: there was no diffusion phenomenon between the experimental group and the matched group for ink, carbon ink, and pencil, and the $\Delta E^*$ of the experimental group were all the better than those of the matched group. In addition, pure blue and red ink still had a more severe diffusion phenomenon, and the $\Delta E^*$ is also enormous but still better than that of the matched group. The reinforcement effect tends to improve with the increase in the concentration of the reinforcing agent. The blue-black inks have no diffusion and the $\Delta E^*$ gradually increased with increasing reinforcing agent concentration. All concentrations were higher than 19.98, except for the data on 3% and 5%, which are smaller than those of the matched group. This phenomenon may be related to the oxidation of the carbon–carbon double bond of the acrylic acid in the reinforcing agent by the solid oxidizing agent $H_2O_2$ resulting in the partial failure of the reinforcing agent. Meanwhile, the residual ferrous tannic acid in the blue-black inks was also oxidized by $H_2O_2$, and the handwriting color changed significantly under the joint influence of both (as shown in Table 5 and Figure 8).

**Table 5.** Diffusion results of oxidation resistance experiment on reinforced handwriting materials.

| Oxidation Resistance of Writing Material | Matched Group | Concentration | | | | |
|---|---|---|---|---|---|---|
| | | 3% | 5% | 10% | 15% | 20% |
| Ink | — | — | — | — | — | — |
| Carbon ink | — | — | — | — | — | — |
| Blue-black ink | — | — | — | — | — | — |
| Pure blue ink | +++ | +++ | +++ | +++ | +++ | +++ |
| Red ink | +++ | +++ | +++ | +++ | +++ | +++ |
| Pencil | — | — | — | — | — | — |

The diffusion phenomenon is represented by these symbols: no noticeable change in handwriting (—); tiny (tiny); mild (+), obvious (++), severe (+++).

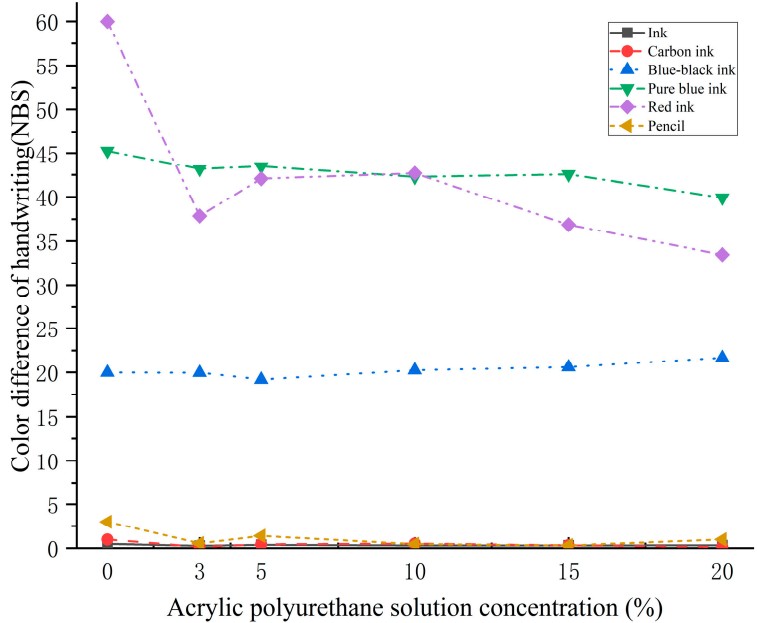

**Figure 8.** Changes in oxidation resistance of handwriting after reinforcement.

(4) Dry heat aging resistance: the experiment was recorded for 36 h and 72 h of dry heat aging data; the data on the experimental group are still better than those of the matched group, which can be seen in Table 6. The ink, carbon ink, and red ink were well reinforced and the $\Delta E^*$ did not fluctuate significantly with increasing heat aging time, which also indicates that the $\Delta E^*$ of the reinforcing agent did not change significantly with time. The $\Delta E^*$ of black ink, pure blue ink, and pencil is relatively large between 36 h and 72 h. Except for ink and pencil, other handwriting materials show the trend of $\Delta E^*$ decreasing proportionally to the increase in concentration.

Through a series of experiments, the anti-fading property of the pigment components in paper cultural relics and the combination of pigment and paper are still the key factors in determining the durability of paper cultural relics. The blue-black ink with pigment as the main component and paper as a combination of absorption demonstrated medium performance in the experiment, however, care must be taken to ensure that it does not come into contact with oxidizing solid substances. The pigment composition of ink and carbon ink is carbon black, which has high stability and demonstrated the best reinforcement effect in this experiment. However, the same pencil with carbon black as the main component performed poorly in the experiment. It showed a poor reinforcement effect in the acid resistance and dry heat resistance aging experiment but performed well in alkaline and oxidation resistance. No specific reason has been found for this phenomenon. Pure blue ink with dye-based components and red ink showed the worst performance in all experiments,

both of which were less than 3, however, it is worth noting that in the dry heat aging experiment the red ink showed a good performance on the whole.

Overall, the experimental group outperformed the matched group after reinforcement with acrylic polyurethane solution and showed some reinforcement performance.

**Table 6.** Experimental results of dry-heat aging of reinforced handwriting materials.

| The Writing Material Dry and Aging | Time | Matched Group | Concentration | | | | |
| --- | --- | --- | --- | --- | --- | --- | --- |
| | | | 3% | 5% | 10% | 15% | 20% |
| Ink | 36 h | 0.95 | 0.29 | 0.32 | 0.29 | 0.2 | 0.23 |
| | 72 h | 1.06 | 0.25 | 0.35 | 0.16 | 0.3 | 0.32 |
| Carbon ink | 36 h | 0.9 | 1.28 | 0.88 | 0.72 | 0.87 | 0.37 |
| | 72 h | 1.12 | 1.39 | 0.95 | 0.75 | 0.9 | 0.41 |
| Blue-black ink | 36 h | 14.49 | 15.44 | 15.43 | 11.85 | 5.15 | 6.24 |
| | 72 h | 19.47 | 18.01 | 19.09 | 18.42 | 12.65 | 8.43 |
| Pure blue ink | 36 h | 15.17 | 16.62 | 12.15 | 10.9 | 4.94 | 5.37 |
| | 72 h | 19.28 | 17.61 | 15.29 | 16.35 | 6.06 | 6.4 |
| Red ink | 36 h | 3.04 | 2.76 | 1.4 | 1.57 | 0.41 | 1.08 |
| | 72 h | 3.24 | 2.74 | 2.09 | 1.67 | 1.6 | 0.88 |
| Pencil | 36 h | 4.62 | 3.35 | 3.01 | 3.75 | 3.43 | 3.94 |
| | 72 h | 6.45 | 5.1 | 5.29 | 5.57 | 5.22 | 5.01 |

*3.5. Appearance and Characterization*

3.5.1. Internal Microstructure Analysis of Paper

A 3D measurement laser microscope characterizes the internal morphology of the paper. Figure 9a,b are laser images of paper: (a) is the surface of unreinforced paper fiber, which shows that there are many fine chapped lines on the fiber surface; (b) is the surface of reinforced paper fiber, and it can be observed that the fiber surface becomes smooth and has no chapped lines, which shows that the fiber surface is covered with a layer of acrylic polyurethane film. In addition, the original loose mesh structure is still maintained among the fibers, and the inherent texture of the paper can be preserved; (c) and (d) are color images of the 3D measurement laser microscope; (c) shows that colored resin cross-linked parts can be observed between fibers; and (d) found more oversized resin embedded between fibers. These increase the strength of paper fibers.

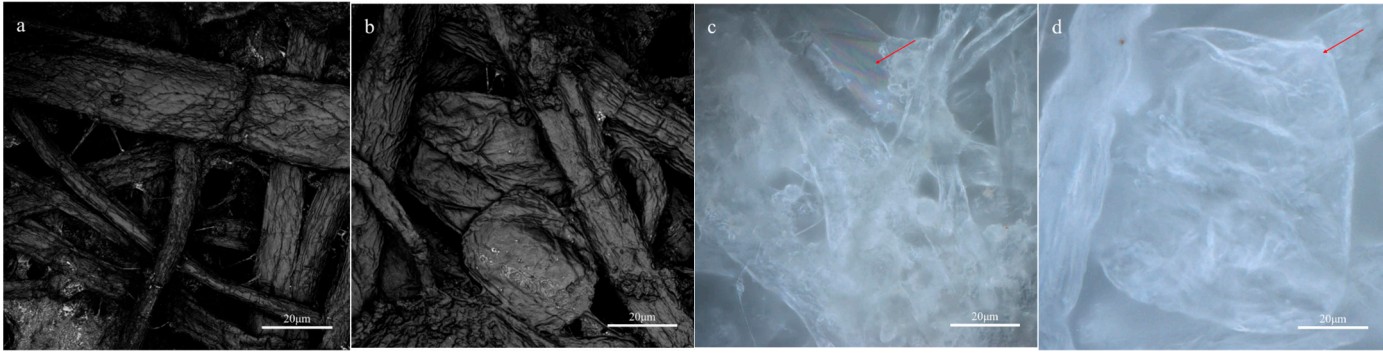

**Figure 9.** The morphology of paper was observed by laser microscope. (**a**) Laser image of unreinforced paper fibers; (**b**) Laser image of reinforced paper fibers; (**c**) Color image of reinforced paper fibers cross-linking; (**d**) Color image of the colloidal particles between reinforced paper fibers.

3.5.2. Micro-Morphology Analysis of Handwriting

To further confirm the protective effect of acrylic polyurethane reinforcement on pigment particles, the handwriting samples after reinforcement were observed by SEM. In (a), it can be seen that the fiber surface is smooth, which is consistent with the observation results in Figure 9b. Figure 10b is the enlarged observation diagram of pigment particles in the red label of (a), which aimed to prove the protective effect of the reinforcement further.

However, it could not be proved in the observation of (b). Therefore, EDS analysis was performed on the unreinforced handwriting sample (c) and reinforced handwriting sample (d). Cellulose comprises C, H, and O elements, and the reinforcement agent contains N elements. Therefore, the content of the N element in the sample is detected to verify that there is a protective film formed by the reinforcement agent on the pigment particles. The experimental results show that the content of the N element in (c) is 0. In contrast, the N element in (d) is significantly increased, indicating that a protective film has been generated over the handwriting.

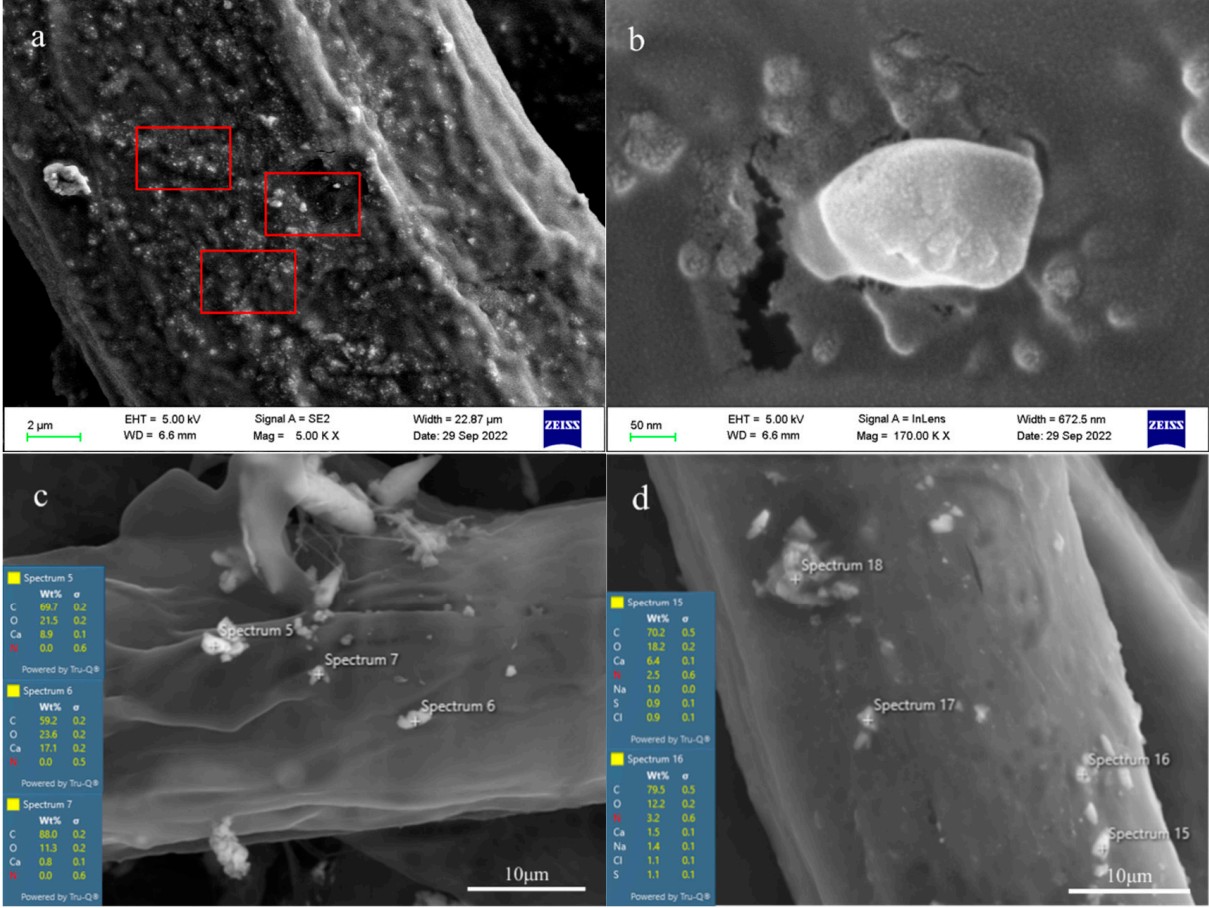

**Figure 10.** SEM and EDS of handwriting. (**a**) SEM image of reinforced paper fibers; (**b**) SEM image of reinforced pigment particles; (**c**) EDS analysis of N element content in unreinforced paper; (**d**) EDS analysis of N element content in reinforced paper.

### 3.5.3. IR Spectrum Analysis after Durability Test

In order to further explore the aging resistance of acrylic polyurethane, infrared spectroscopy analysis was conducted on the paper samples after durability experiments. Use the reinforced paper samples as the matched group. In Figure 11, 3326 cm$^{-1}$ is the characteristic peak of the paper -OH group and 2904 cm$^{-1}$ is the characteristic peak of cellulose methyl and methylene. The characteristic peak of the unreacted -NCO group in acrylic polyurethane is 2268 cm$^{-1}$. The characteristic peak of carbamate C=O produced by the reaction of reinforcing agent and paper hydroxyl is 1682 cm$^{-1}$. The characteristic peak of C-N in acrylic polyurethane is 1026 cm$^{-1}$. It can be seen that the characteristic peaks of -NCO, C=O, and C-N that appeared in the matched group remained after the durability experiment, and the height of the characteristic peaks changed little, indicating that acrylic polyurethane has good durability.

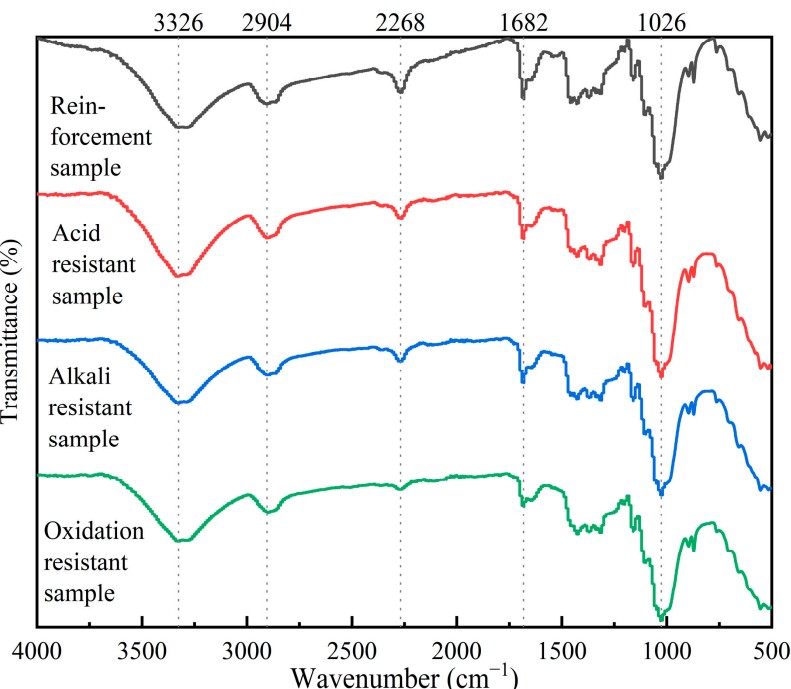

**Figure 11.** Sample IR spectrum after durability test.

## 4. Conclusions

This article aimed to study the effect of acrylic polyurethane materials on the handwriting of paper cultural relics. Reinforcement experiments reinforced six types of handwriting materials by preparing acrylic polyurethane solutions with different concentrations and exploring the properties of acrylic polyurethane through durability experiments, microstructure observation, and energy spectrum analysis. The authors draw the following conclusions:

(1) It can be seen from the experimental data that the writing reinforced by acrylic polyurethane solution has no diffusion. In the durability test, the diffusion degree of the experimental group is significantly smaller than that of the matched group. In addition, we can obtain the best reinforcement concentration of different handwriting materials. The best reinforcement concentration for ink is within 15% and within 10% for carbon ink. The reinforcement concentration for blue and black ink is 20%, 15% to 20% for pure blue ink, and 3% for red ink and pencil. In practical application, the characteristics of the writing material should be considered, and the appropriate reinforcement concentration can achieve the ideal reinforcement effect.

(2) According to laser microscopy and SEM, the surface of the reinforced paper fibers is smooth. The formation of larger resin particles between fibers enhances the strength of paper fibers. According to EDS analysis, there is also an acrylic polyurethane coating on the surface of the pigment particles. The IR detection after the durability experiment shows that acrylic polyurethane has excellent durability performance. Overall, acrylic polyurethane materials are effective in protecting writing materials.

**Author Contributions:** Writing—original draft preparation, T.M. and X.L. (Xufeng Li); background research, X.S.; project administration, Y.H.; data curation, J.Z.; writing—review and editing, X.L. (Xueke Luo) and Y.C.; funding acquisition, Y.C. All authors have read and agreed to the published version of the manuscript.

**Funding:** This research was funded by the National Key Research and Development Program of China (No. 2021YFD1600402), the Key Research and Development Plan of Shaanxi Province (Grant No. 2020GXLH-Z-031), the Key Scientific Research Project of Weinan City (Grant No. 2020ZDYF-GYCX-107), the Key Scientific Research Project of Education Department of Shaanxi Province (Grant No. 20JS106), Central Guidance on Local Science and Technology Development Fund of Shaanxi Province (No. 2020-ZYYD-NCC-9), Shaanxi Provincial Department of Education Collaborative Innovation Center Project (20JY052), National Natural Science Foundation of China (No. 52072075), Science and Technology Planning Project of Qindu District, Xianyang City (No. 2021QKJ-001), Outstanding Youth Science Fund of Shaanxi Province (No. 2018JC-028).

**Institutional Review Board Statement:** Not applicable.

**Informed Consent Statement:** Not applicable.

**Data Availability Statement:** No data were created or analyzed in this study. Data sharing does not apply to this article.

**Conflicts of Interest:** The funders had no role in the design of the study; in the collection, analyses, or interpretation of data; in the writing of the manuscript; or in the decision to publish the results.

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
