# Peer review of "Study on the Performance of Acrylic Polyurethane for the Protection of Handwriting on Paper Relics"

_coatings, doi:10.3390/coatings13050822_

Round 1

Reviewer 1 Report

The topic you are working on is very interesting and relevant.

I am recommending a rejection of this article in current form. Please find below some suggestions to improve the article.

Major issues

1.     Language: throughout the article, there are many mistakes. There is a need for language editing. Take the help of some professional language editor or services. In this case you should provide the certificate for language editing.

2.     Research topic is clearly defined, though more deepened literature review dealing with protection and restoration of handwriting on paper is missing.

3.     Appropriate methods for characterization of performance of coated paper were chosen. In description of methods used for performance testing the reference to the standard is missing. Authors should explain why they chose 0.01 mol/L solution and exposure time of 1 hour.

4. Table 2 and Table 6: Description is wrong

5. Results of the measurements of color change after different treatments are displayed twice, once in a table, and the second time as an figure (Table 2 and Figure 3, Table 3 and Figure 4, Table 4 and Figure 5, Table 5 and Figure 6). This is an unnecessary duplication. 

Reviewer 2 Report

Article "Study on the performance of acrylic polyurethane for the protection of handwriting on paper relics" by Mao et al. 

Although the paper could be of interest for readers, some important issues should be considered: 

- Introduction (Line 28-29): Please explain more profusely why handwritting is one of the most important factors determining the life of paper.

- Experimental Materials and Methods (Lines 51-56): Data of equipments must be completed. Most of them are missing, especially colorimeter, IR spectrometer and SEM)

- Lines 61-62: Differences between inks?

- Lines 98-100: CIEL*a*b*, a*, b*

- Line 110: Why you consider the minium value of 1.5 for delta E*?

- Results and Discussion:

Figure 2: Legend should be more explicative

Tables: diffusion is quantified visually? what is the meaning of slight, -, +, +++?

Regarding acid resistance, please show the infrared spectra after attack. The same for alkali attack and for H2O2 attack. 

Please specify the mode of quantification in EDS analyses? N amounts?

Figure 8: secondary and in-lens images? You should specify the detectors employed. 

Reviewer 3 Report

In this Manuscript, the authors studied the Study on the Performance of Acrylic Polyurethane for the Pro-tection of Handwriting on Paper Relics. The topic of this paper is interesting and useful. Therefore, this Manuscript could be acceptable for publication but it needs some revisions to help the authors and ensure the quality of the published papers in this journal.

1. The whole manuscript must be carefully revised for sentence construction and language errors. 

2. The important results must be described in abstract. For example, acid resistance, alkali resistance, oxidation resistance, dry heat ageing resistance and durability realizing based on the concentrations.

3. Please update your introduction with more related articles such as :

Xu, Jicheng, et al. "Preparation of self‐healing acrylic copolymer composite coatings for application in protection of paper cultural relics." Polymer Engineering & Science 60.2 (2020): 288-296.

4.  The topic has been studied in the past and much is already known about the behavior of these polymer matrices in acid, alkali, oxidation, dry heat againg resistance. Therefore, the authors need to make a stronger argument for the novelty of this work and point out some obvious new contributions to the materials science literatures. It needs to be better sold as novel contributions. 

5.   Be consistence either Fig. or Fig

6. The authors stated that Pigment particles on the fiber surface and the surface of the fibers remained smooth which can be seen in (a). Nevertheless, no significant film formation was observed after magnification of the pigment particles in (b). This is not clear from SEM images Also please explain this statement: on the fiber surface and the surface of the fibers???

7. The conclusions section does not really excite the reader that the work presented in this study is new and interesting – the conclusion should be improved, please rewrite it with complying with the abstract.

8. It would be helpful to have a TEM images showing the composites materials.

9. Please include the size and shape of the samples for all measurements.

Round 2

Reviewer 1 Report

Dear authors,

you have improved manuscript substantially. All my proposed suggestions were considered. I have no further requirements.  

Reviewer 2 Report

-